# Cysteine Residues in Region 6 of the *Plasmodium yoelii* Erythrocyte-Binding-like Ligand That Are Related to Its Localization and the Course of Infection

**DOI:** 10.3390/biom13030458

**Published:** 2023-03-02

**Authors:** Hitoshi Otsuki, Osamu Kaneko, Daisuke Ito, Yoko Kondo, Hideyuki Iriko, Tomoko Ishino, Mayumi Tachibana, Takafumi Tsuboi, Motomi Torii

**Affiliations:** 1Division of Medical Zoology, Department of Microbiology and Immunology, Faculty of Medicine, Tottori University, Yonago 683-8503, Japan; 2Department of Protozoology, Institute of Tropical Medicine (NEKKEN), Nagasaki University, Nagasaki 852-8523, Japan; 3Division of Global Infectious Diseases, Department of Public Health, Kobe University Graduate School of Health Sciences, Kobe 654-0142, Japan; 4Department of Parasitology and Tropical Medicine, Graduate School of Medical and Dental Sciences, Tokyo Medical and Dental University, Tokyo 113-8510, Japan; 5Division of Molecular Parasitology, Proteo-Science Center, Ehime University, Toon 791-0295, Japan; 6Division of Cell-Free Sciences, Proteo-Science Center, Ehime University, Matsuyama 790-8577, Japan

**Keywords:** erythrocyte, invasion, malaria, trafficking, ligand

## Abstract

*Plasmodium* malaria parasites use erythrocyte-binding-like (EBL) ligands to invade erythrocytes in their vertebrate host. EBLs are released from micronemes, which are secretory organelles located at the merozoite apical end and bind to erythrocyte surface receptors. Because of their essential nature, EBLs have been studied as vaccine candidates, such as the *Plasmodium vivax* Duffy binding protein. Previously, we showed through using the rodent malaria parasite *Plasmodium yoelii* that a single amino acid substitution within the EBL C-terminal Cys-rich domain (region 6) caused mislocalization of this molecule and resulted in alteration of the infection course and virulence between the non-lethal 17X and lethal 17XL strains. In the present study, we generated a panel of transgenic *P. yoelii* lines in which seven of the eight conserved Cys residues in EBL region 6 were independently substituted to Ala residues to observe the consequence of these substitutions with respect to EBL localization, the infection course, and virulence. Five out of seven transgenic lines showed EBL mislocalizations and higher parasitemias. Among them, three showed increased virulence, whereas the other two did not kill the infected mice. The remaining two transgenic lines showed low parasitemias similar to their parental 17X strain, and their EBL localizations did not change. The results indicate the importance of Cys residues in EBL region 6 for EBL localization, parasite infection course, and virulence and suggest an association between EBL localization and the parasite infection course.

## 1. Introduction

Malaria is caused by the infection of *Plasmodium* parasites, and the disease is estimated to have killed 619,000 people in 2021 [1]. The most virulent species of malaria parasite, *Plasmodium falciparum*, is mainly distributed in tropical and sub-tropical regions of Sub-Saharan Africa. A less-virulent species, *Plasmodium vivax*, shows a wider global distribution in tropical areas and temperate zones. The malaria parasite displays virulence during the human blood stage due to an amplifying cycle of erythrocyte invasion by the merozoite stage, intraerythrocytic development, and ultimately lysis of the host erythrocytes. Understanding the mechanism of erythrocyte invasion by merozoites is a critical step to preventing malaria disease and death. Recognition and invasion of erythrocytes is mediated by a number of adhesive ligand molecules which are secreted from invasion-related organelles called micronemes, rhoptries, and dense granules [2]. Several steps are involved in the invasion process, and some finer details have recently been revealed by live-imaging technology [3]. Essential steps to invade the erythrocyte include a re-orientation of the *Plasmodium* merozoite on the erythrocyte surface and the formation of an irreversible tight junction between the parasite and the host cell. Erythrocyte-binding-like (EBL) proteins secreted from micronemes are involved in tight junction formation; and a lack of the erythrocyte receptor or the disruption of the gene locus coding for EBL for *Plasmodium knowlesi* or *Plasmodium yoelii*, respectively, resulted in failed cell invasion without forming a tight junction [4,5,6]. This is explained by the experimental demonstration that *P. vivax* EBL, termed the Duffy binding protein (PvDBP), recognizes the Duffy antigen and *P. vivax* cannot invade erythrocytes lacking the Duffy antigen [7,8]. The Duffy antigen is a minor blood type antigen and chemokine receptor on the erythrocyte and serves to modulate chemokine concentrations in the blood stream. Most Sub-Saharan Africans do not express the Duffy antigen on their erythrocytes and are resistant to *P. vivax* infection, resulting in the scarce distribution of *P. vivax* in Sub-Saharan Africa [4].

Because of its essentiality, PvDBP has been studied as a target of vaccine development aiming to prevent parasite invasion of erythrocytes. EBL is a type I transmembrane protein with distinct cysteine-rich domains at the N-terminal and C-terminal sides of the extracellular region. The N-terminal Cys-rich domain binds erythrocyte surface receptor molecules and is called region 2. This region has been extensively studied in vaccine development for *P. vivax* PvDBP and the *P. falciparum* ortholog EBA-175 [9,10]. Region 2 has also been studied for its role in erythrocyte invasion, such as conferring the binding ability to recognize erythrocyte surface receptor molecules [4,11,12,13]. In contrast, studies on the C-terminal cysteine-rich domain, termed region 6, are relatively limited (see Figure 1A as a representative primary structure of EBL).

Unlike human and simian malaria parasites, EBL molecules in rodent malaria parasites are not represented by a multigene family. Therefore, the rodent malaria model, such as *P. yoelii*, provides a platform in which EBL function can be analyzed without interference from paralogous molecules [14]. EBL region 6 has eight Cys residues which are conserved among known functional EBL molecules of *Plasmodium* spp. The *P. yoelii* 17X strain shows low parasitemia and a preference to invade immature erythrocytes. The 17XL strain was derived from the 17X strain following passage in mice and in contrast, it shows significantly higher parasitemia, invasive ability into all erythrocytes, and lethality in mice [15,16]. We found that the *P. yoelii* 17X non-lethal strain contains eight Cys residues in PyEBL region 6; whereas the 2nd Cys within the domain is substituted to Arg in the 17XL lethal strain. In addition, we found that the PyEBL protein is located on the dense granules in the 17XL strain, in contrast to a micronemal localization for all known EBLs possessing eight intact Cys residues in region 6 [17]. Our allelic exchange experiments revealed that the Cys to Arg substitution in the 17XL strain was responsible for altering the parasite infection course, virulence, and PyEBL localization [17].

In vitro continuous culture of *P. vivax* is not established, and in vivo animal experiments are rarely performed because they require a primate system. *P. vivax* is phylogenetically more closely related to *P. yoelii* than to *P. falciparum*. Both *P. yoelii* and *P. vivax* have single EBL genes, whereas *P. falciparum* has multiple EBL members. *P. vivax* prefers to invade reticulocytes, as do *P. yoelii* 17X parasites. Therefore, we think that findings obtained from the *P. yoelii* rodent malaria model are likely to contribute to understanding PvDBP function in *P. vivax*.

Treeck et al. (2006) reported that *P. falciparum* EBA-175 region 6 was responsible for its trafficking to the micronemes [18]. The crystal structure of EBA-175 region 6 suggested the disulfide bonding pattern of eight Cys residues [19]. However, it remains unclear if other Cys residues in PyEBL region 6 are also responsible for correct PyEBL localization in the micronemes, and if different PyEBL localization patterns are associated with parasite virulence. Region 6 amino acids are highly conserved among *Plasmodium* parasites and this suggests a functional importance for this domain. The eight cysteines likely form four disulfide bonds which underpin the domain stability and conformation. Functional analysis via comprehensive modification of the disulfide bonds within the domain, and the impact of the modifications on intracellular localization, may cast light on the importance and uniqueness of this vaccine candidate molecule. In this study, we tried to generate transgenic *P. yoelii* lines from a 17X strain parasite, in which all eight Cys residues in PyEBL region 6 are independently replaced with Ala residues to understand the consequence of the replacement of each Cys residue for the parasite infection course, virulence, PyEBL localization, and possible associations of these phenotypes. These findings may contribute to vaccine studies by understanding the behavior of vaccine target molecules during the invasion step.

## 2. Materials and Methods

### 2.1. Parasites and Animals

Rodent malaria parasites *P. yoelii* 17X and 17XL strains were cloned by limiting dilution, passed through *Anopheles stephensi*, and infected into BALB/c mice, and the infected blood was frozen with 20% DMSO in liquid nitrogen. ICR, BALB/c mice and Wistar rats (Japan SLC, Hamamatsu, Japan) were maintained at the Research Center for Bioscience and Technology, Tottori University. All experiments were approved and performed in accordance with the guidelines of the Institutional Animal Care and Use Committee of Tottori University. All mice and rats at Tottori University were housed in secured constant temperature and air-conditioned rooms with constant day/night light cycles. The animals were kept in uncrowded cages with regularly exchanged bedding to reduce stress and provided sufficient food and clean water. All the animals were euthanized under CO_2_ gas. The guidelines of animal experiments at Tottori University are publicly disclosed on the following website: (https://www.tottori-u.ac.jp/kouhou/kisokusyuu/reiki_honbun/u095RG00000581.html, accessed on 27 February 2023). These rules comply with Japanese laws and guidelines regulating laboratory animals listed as follows: http://www.env.go.jp/nature/dobutsu/aigo/2_data/laws/nt_h25_84_en.pdf, accessed on 27 February 2023, https://www.scj.go.jp/ja/info/kohyo/pdf/kohyo-20-k16-2e.pdf, accessed on 27 February 2023, and https://www.env.go.jp/nature/dobutsu/aigo/2_data/laws/shobun_en.pdf, accessed on 27 February 2023. The rabbits were maintained at Kitayama Labes Co., LTD. (Ina, Japan) according to their Ethical Guidelines for Animal Experiments.

### 2.2. Plasmid Constructs

The previously described plasmids pPbDT3U-B12 and pHDEF1-mh-R12 were modified and used [17] and are elaborated upon in Appendix A. Briefly, the GFPm2 sequence was amplified using KOD Plus DNA polymerase (TOYOBO, Japan) with primers P1 (5′ ggatccAGTAAAGGAGAAGAACTTTTCAC 3′) and P2 (5′ gtcgacCTATTTGTATAGTTCATCCATGC 3′) at an annealing temperature of 49 °C for 40 cycles; from a pHDEF1-mh-based plasmid [20] containing GFPm2 derived from a plasmid pHRPGFPm2 [21] and ligated into the pGEM-T Easy plasmid (Promega, Madison, WI, USA). The insert was then digested with BamHI and SalI, purified, and ligated into pPbDT3U-B12 to produce pGFP-DT3U-B12. Fragments containing PyEBL region 2 to the cytoplasmic tail (Cyt) and PyEBL 3′ UTR were amplified using primers P3 (5′ gagactcgagTCTTCTGTTAAACCCAGTAATAC 3′) and P4 (5′ tctagaATAAAAATCTACAGGTATATATTC 3′) at an annealing temperature of 60 °C for 40 cycles, and P5 (5′ ccatggCAAAATATTGAATTGAAGCC 3′) and P6 (5′ ctcgagCATGTAATAAATAAATTAATA 3′) at an annealing temperature of 55 °C for 40 cycles. Amplified fragments were subcloned, and the sequences were confirmed. The subcloned fragments and pGFP-DT3U-B12 plasmid were then digested and ligated using T4 DNA ligase to produce donor vector pGFPDT-PyEBLR6mod. The donor vector was mixed with pDONR221 and BP clonase (Invitrogen, Carlsbad, CA, USA), and the reaction was carried out to generate the entry vector pENTRY-PyEBLR6mod. To insert nucleotide mutations conferring single amino acid substitutions into the entry vector, the QuikChange^®^ II Site-Directed Mutagenesis Kit (Stratagene, La Jolla, CA, USA) or PrimeStar^®^ Mutagenesis Basal Kit (TAKARA, Otsu, Japan) and specific primers were used for substitution as shown in Appendix A. All of the mutated entry vectors were validated by sequencing. The PyEBL region 6 mutated entry vector was mixed with pHDEF1-mh-R12 destination vector as described [17], and an LR reaction was performed to generate the pYEBL-PyEBLR6mod-GFP constructs. For transfection the constructs were linearized by digestion with XhoI.

### 2.3. Transfection of the P. yoelii Parasite

A *P. yoelii* 17X schizont-enriched fraction was collected by differential centrifugation on 50% Histodenz (Sigma-Aldrich, St. Louis, MO, USA) in Tris buffer (at 1190× *g*, slow start, and minimum brake); and 25 µg of XhoI-digested transfection constructs was electroporated using the Nucleofector™ device (Amaxa GmbH, Köln, Germany) with Human T cell solution under program U-33 [17] and are elaborated upon in Appendix A. Transfected parasites were intravenously injected into 8- to 10-week-old ICR female mice, which were treated with 0.07 mg/mL pyrimethamine in their drinking water. Reappeared parasites were observed by tail blood smears on glass slides stained with Giemsa staining; blood was collected by cardiac puncture. Drug-resistant parasites were cloned by limiting dilution, specifically, parasite-infected blood was obtained from donor ICR mice infected with transfected parasites under drug pressure and injected at dilutions of one infected erythrocyte per each mouse intravenously for eight naïve ICR mice. The progress of infections was observed with thin blood smears as above, and the infected mice were sacrificed to collect blood by cardiac puncture and preserved. The integrations of the transfection constructs were confirmed by PCR amplification using a set of primers unique for the modified *pyebl* gene locus, followed by sequencing [17].

### 2.4. Parasite Challenge

Donor ICR mice were infected with parasites intraperitoneally. The number of erythrocytes was counted using a hemocytometer and the parasitized erythrocytes observed in Giemsa-stained thin blood smears were measured to calculate the parasitemia. Five or six BALB/c mice aged 8 to 10 weeks were infected intravenously with 1 × 10^4^ parasitized erythrocytes. Giemsa-stained thin blood smears were examined under a light microscope, and parasitemias were recorded daily from days 3 to 7. The parasitemias between the groups were compared by one-way ANOVA with Dunnett’s multiple comparison test implemented in GraphPad Prism 6.0 software (GraphPad software, San Diego, CA, USA). Mouse survival was monitored until day 14. Survival curves were compared by the log-rank (Mantel–Cox) test implemented in GraphPad Prism 6.0 software.

### 2.5. Selectivity Index

To compare erythrocyte preference between Cys-substituted *P. yoelii* 17X transgenic parasites, a selectivity index (SI) was calculated as described [22]. Briefly, the number of observed multiple-infected erythrocytes (containing more than one parasite per erythrocyte) was divided by the expected number of multiple-infected erythrocytes according to a random Poisson distribution, which was calculated from the number of infected erythrocytes and parasitemia [22]. When the preferred erythrocyte type is limited, SI shows a higher value. More than 200 parasitized erythrocytes were examined on Giemsa-stained thin blood smears collected on day 4 post infection. The SI of each group was compared by the one-way ANOVA test with Tukey’s multiple comparisons test implemented in GraphPad Prism 6.0.

### 2.6. Antibody Production

All recombinant proteins for immunization were expressed using the wheat germ cell-free protein synthesis system as described [17,23] and are elaborated upon in Appendix A. To produce rat anti-PyEBL sera, female Wistar rats (Japan SLC) were intraperitoneally immunized three times at three-week intervals with 75 µg of recombinant PyEBL R1-6 emulsified with Freund’s adjuvant and sacrificed 14 days after the last immunization for serum collection. The generation of anti-PyEBL monoclonal antibody 1G10 recognizing region 4 was described [17]. To produce rabbit anti-PyAMA-1 serum, a gene fragment of *P. yoelii* AMA-1 (PyAMA-1) encoding the ectodomain of PyAMA-1 (C21—K479) was amplified with primers (5′ gagagagactcgagTCCGAAGGTCCAAATCAAGTTATTTC 3′) and (5′ gagagagaggatccTCATTTCTGTTTTGGGTTTTCATAGTCAC 3′) at 55 °C for 35 cycles with KOD plus polymerase (the XhoI and BamHI restriction sites are underlined.). The amplicon was cloned into the pEU-E01-GST vector designed for the wheat germ cell-free system (CellFree Sciences, Matsuyama, Japan). Then, the recombinant PyAMA-1 protein was expressed using the wheat germ cell-free system (CellFree Sciences) as a GST fusion protein, affinity-purified using a glutathione-Sepharose 4B column (GE Healthcare, Bio-Sciences, Piscataway, NJ, USA) and cleaved with TEV protease (Invitrogen) to remove the GST-tag as described [23]. A female Japanese white rabbit was subcutaneously immunized three times with 250 µg of recombinant PyAMA-1 emulsified with Freund’s adjuvant at three-week intervals.

### 2.7. Immunofluorescence Microscopy

Transfected parasite-infected blood was taken from the infected mice and washed twice with 4 °C phosphate-buffered saline (PBS) by centrifugation at 400× *g* for three minutes. Thin blood smears were air-dried and stored at −70 °C for the preservation of their antigenicity. Thawed smears were fixed with ice-cold acetone and blocked with 5% skim milk (Wako, Osaka, Japan) in PBS (blocking buffer) for 30 min at room temperature before immunostaining. The blocking buffer was removed by aspiration, and then antibodies were diluted 100 times with blocking buffer and incubated at 37 °C for one hour. The antibodies were aspirated and the slides were soaked in ice-cold PBS for five minutes. After washing, the smears were stained at 37 °C for 30 min with Alexa-488 conjugated goat anti-mouse IgG and Alexa-555 conjugated goat anti-rabbit IgG secondary antibodies (Molecular Probes, Eugene, OR, USA) diluted 500-fold with blocking buffer. Parasite nuclei were stained with Hoechst 33342 (Thermo Fisher Scientific, Waltham, MA, USA). Images were captured with an LSM780 laser confocal inverted microscope (Zeiss AG, Oberkochen, Germany). Co-localization analysis of PyEBL and PyAMA-1 was carried out on 15 mature merozoites from each parasite line, and the data were analyzed with ZEN 2011 software (Zeiss) and correlation coefficient R values were obtained. The R values of each parasite line were compared by the one-way ANOVA test with Dunnett’s multiple comparisons test implemented in GraphPad Prism 6.0. Live transgenic parasites were also observed to confirm a GFP signal using a BX60 fluorescence microscope (Olympus, Tokyo, Japan). Five µL of tail blood was taken from an infected mouse and diluted at room temperature in 100 µL of PBS containing Hoechst 33342 diluted 1000-fold. Photographs were taken with a DS-Ri1 digital camera (Nikon, Tokyo, Japan).

### 2.8. Immunoelectron Microscopy

Schizonts from transgenic lines were enriched by differential centrifugation on 50% Histodenz in Tris buffer as described in Section 2.3 and fixed in 1% paraformaldehyde and 0.1% glutaraldehyde in HEPES-buffered saline (pH 7.05) and embedded in LR white resin (Polysciences, Warrington, PA, USA). Sections were blocked for 30 min in PBS containing 5% skim milk and Tween 20 (PBS-milk-Tween 20), incubated overnight at 4 °C in PBS-milk-Tween 20 containing rat anti-PyEBL R1-6 serum (1:50 dilution), and then incubated at 37 °C for 1 h in PBS-milk-Tween 20 containing goat anti-mouse IgG conjugated with gold particles (15 nm diameter; Jansen, Piscataway, NJ, USA) diluted 1:20 in PBS-milk-Tween 20. The sections were stained with 2% uranyl acetate in 50% methanol in DDW and examined by electron microscopy (JEM-1230; JEOL, Tokyo, Japan).

## 3. Results

### 3.1. Seven of the Eight Cys Residues in PyEBL Region 6 Were Successfully Replaced with Ala

PyEBL region 6 contains eight Cys residues, and the number and positions are conserved across the genus *Plasmodium*. The 3D structure analysis of recombinant *P. falciparum* EBA-175 region 6 revealed that these eight Cys residues participate in four disulfide bonds within the domain (Figure 1) [19]. We have reported that in the lethal 17XL strain, the 2nd Cys residue at amino acid position 726 in EBL region 6 is substituted to Arg (C726R) [17]. To investigate the importance of the Cys residues contributing to disulfide bond formation in the native parasite protein, we tried to independently substitute all eight Cys residues in PyEBL region 6 to Ala residues, an amino acid chosen for its small size and neutrality. For localization analysis a GFP tag was fused at the C-terminus of the EBL protein. Transgenic parasite lines were obtained substituting seven Cys, with the exception of the 6th Cys at position 754; all transgenic lines showed a GFP signal (Appendix A). Sequencing analysis of the transgenic lines confirmed that the PyEBL gene locus was modified as designed.

Seven Cys residues were successfully substituted, whereas the 6th residue C754 was refractory to Ala substitution, despite four transfection attempts to obtain a transgenic C754A line and one attempt at replacement to C754R. The control plasmid with the same backbone sequence to substitute from TGT to TGC, both of which encode Cys, was successful. The predicted disulfide bond partner of C754 was successfully substituted to Ala, at amino acid position 768, indicating that the non-viability of C754A transfectants was not due to the loss of the disulfide bond formed between the 6th and 7th Cys residues. These results are summarized in Appendix A.

### 3.2. Effect on the Infection Course by Substituting Cys to Ala in PyEBL Region 6

The *P. yoelii* 17X parasite shows a non-lethal phenotype with low parasitemia and a preference to invade reticulocytes, whilst the 17XL lethal strain shows a higher parasitemia of more than 80% and no preference for a target erythrocyte type. Our previously reported transgenic 17X parasite in which the 2nd Cys in PyEBL region 6 was substituted to Arg also showed a higher parasitemia, almost as high as the lethal 17XL strain [17]. To investigate in further detail, we examined the infection courses of seven established transgenic parasite lines up to day 7 post infection. Figure 2A,B shows that the infection course and parasitemia on day 6 can be roughly classified into two groups. One group contains the C717A, C726A, C747A, C748A, and C768A lines and showed higher parasitemias than the 17X strain, a similar pattern to 17XL. The other group consists of the C738A and C780A lines, with them growing slightly slower and leading to lower parasitemia on day 6 than the parental 17X strain, although the difference was not significant (Figure 2A,B).

### 3.3. Effect on Mouse Survivability by Substituting Cys to Ala in PyEBL Region 6

Virulence was evaluated as the survivability of the infected mice. Infection with the parental 17X strain did not kill any mice, whereas the lethal 17XL strain parasite killed all the infected mice by day 9. The survival of mice infected with PyEBL region 6 transgenic parasites was classified into two distinct groups. One is the sublethal group, including the C726A, C747A, and C748A lines, which killed more than two mice but not all (Figure 2C). These lines showed similar levels of parasitemia as the lethal 17XL strain and were significantly higher than the other transgenic lines (Figure 2B). The second group did not kill any mice by day 9; specifically, the C717A and C768A lines with higher parasitemia and the C738A and C780A lines (Figure 2B,C). A significant difference in the survival curves was detected by the log-rank test (*p* < 0.0001).

### 3.4. Effect on Erythrocyte Preference by Substituting Cys to Ala in PyEBL Region 6

To investigate whether the altered infection courses observed in the transgenic lines were caused by a difference of erythrocyte preference in the invasion process, a selectivity index (SI) was calculated by multiple parasite infections of single erythrocytes for each parasite line on day 4 post infection [22,24] (Table 1). A higher SI value indicates that the parasites invaded into a limited population of erythrocytes, and lower SI means that the parasites invaded into a broader population of erythrocytes. We found that five lines with higher parasitemia (C717A, C726A, C747A, C748A, and C768A) showed low SI (0.5~7.3), which is similar to the lethal 17XL strain (SI = 6.8) and lower than the parental 17X strain (SI = 14.6, no significance). This suggests that they invaded a wider range of erythrocyte populations than the 17X strain. In contrast, the C738A and C780A lines showed significantly higher SI (46.8 and 37.3) than the parental 17X strain (*p* < 0.01) suggesting that they preferred to invade into a more limited population of erythrocytes than the non-lethal 17X strain, which is consistent with the slower growth of these two transgenic lines than the 17X strain.

### 3.5. Effect on the PyEBL Localization by Substituting Cys to Ala in PyEBL Region 6

In our previous study, the parasite infectivity and erythrocyte preference of *P. yoelii* 17X lineages were related to the localization of PyEBL; specifically, the 17X-type showed microneme localization whereas the 17XL-type showed dense granule localization [17]. To investigate the relationship between PyEBL localization and parasite infectivity, IFA was carried out on mature merozoites in schizonts for each parasite line. PyAMA-1 was used as a microneme marker at the merozoite apical end. The C738A and C780A lines with lower parasitemia showed a clear punctate pattern that largely overlapped PyAMA-1. In contrast, the other five lines with high parasitemia showed a somewhat dispersed pattern and were not well colocalized with PyAMA-1 (Figure 3A). Next, we quantitatively assessed the co-localization of PyEBL and PyAMA-1 signals by calculating the correlation coefficient value R for each of the transgenic lines. The R values of five lines (C717A, C726A, C747A, C748A, and C768A) were approximately zero suggesting that the location of PyEBL and PyAMA-1 are different. In contrast, the values of the C738A (3rd Cys) and C780A (8th Cys) lines were more than 0.25 suggesting that there is relatively close localization between PyEBL and PyAMA-1 in these two lines (Figure 3B). These two Cys residues are reported to form a disulfide bond in the EBA-175 region 6 structure [19]. Thus, these results suggest that disruption of one of the four disulfide bonds in PyEBL region 6 does not affect the PyEBL localization in the micronemes.

Immunoelectron microscopy was carried out to further confirm the localization of PyEBL in the merozoite. In merozoites obtained from the C738A and C780A lines, which showed lower parasitemia, gold particle deposition indicating EBL localization was observed on the micronemes (Figure 4). In the other five transgenic parasite lines that showed higher parasitemia, no deposition of gold particles was detected in either the organelles or cytoplasm of merozoites (Appendix A). This is in contrast to our previous study, in which the lethal 17XL strain parasite showed PyEBL localization on dense granules after staining with mouse anti-PyEBL R1-6 antibodies.

## 4. Discussion

We previously reported that a lethal rodent malaria parasite *P. yoelii* 17XL strain had only one amino acid substitution in PyEBL region 6 (R726) compared with the whole PyEBL sequence from the non-lethal *P. yoelii* 17X strain (C726). This caused a significant increase in infectivity to erythrocytes and an altered localization of PyEBL from micronemes to dense granules [17]. In this study, we hypothesized that a stable conformation of Cys-rich PyEBL region 6 is dependent on the four disulfide bonds within this region, and alteration of this integrity impacts PyEBL localization to the micronemes and the parasite infectivity in *P. yoelii* 17X. We succeeded in generating seven transgenic parasite lines in which Cys residues are substituted to Ala and demonstrated that disruption of three out of four disulfide bonds in PyEBL region 6 is associated with phenotypic changes in infectivity and PyEBL localization.

Two transgenic parasite lines, C738A and C780A, did not exhibit a change in the PyEBL localization from micronemes and did not increase their infectivity; rather, they grew slower than the parental 17X strain although the difference was not significant. These results suggest that the predicted disulfide bond between Cys residues at amino acid positions 738 (3rd Cys) and 780 (8th Cys) is supported in the parasite native protein and is not essential for the trafficking of PyEBL to the micronemes. This is in agreement with the interpretation of EBA-175 region 6′s structure; specifically, that this disulfide bond is the connection outside the important protein core and contributes to maintaining the structural stability of the EBA-175 region 6 [19]. However, the slight reduction in the growth rate of these two lines may possibly suggest that the disruption of the disulfide bond between C738 and C780 affects one or more downstream events after trafficking to the micronemes, for example, secretion of PyEBL from the micronemes onto the merozoite surface or recognition of the erythrocyte receptor.

Five transgenic parasite lines showed significantly higher parasitemias than the parental 17X strain, namely, C717A, C726A, C747A, C748A, and C768A. In these lines PyEBL appears to not localize in the micronemes, similar to the lethal 17XL strain which contains a C726R substitution. These Cys residues are predicted to be involved in three disulfide bonds which are suggested to be responsible for the structural stability of the protein core of EBA-175 region 6: C717–C748, C726–C747, and C754–C768 (Figure 1A) [19]. Our findings suggest that the loss of one of these three disulfide bonds in PyEBL region 6 alters the protein structure and as a consequence, PyEBL is not trafficked to the microneme, leading to enhanced parasite infectivity. Two groups have independently conducted linkage group selection analysis to identify genes that determine differences in the parasite multiplication rate of different *P. yoelii* strains [25,26]. Both groups identified the strongest association of the *pyebl* gene locus on chromosome 13 with the virulence of the parasites. In the first study [25], the lethal *P. yoelii* YM strain was crossed with the non-lethal 33XC strain and the substitution of the second Cys to Arg of PyEBL region 6 in the YM strain was identified as a determinant, which is identical to the substitution found in the 17XL strain [17]. The second study conducted a genetic cross between the lethal *P. yoelii nigeriensis* N67 and the non-lethal *P. yoelii* 17XNL strain and identified that the 6th Cys of PyEBL region 6 was substituted to Tyr in the N67 strain [26]. The single amino acid substitutions associated with higher virulence are predicted to have similar disruptive effects on PyEBL region 6, specifically, C726R in 17XL [17], C713R in YM [25], and C741Y in N67 [26]. These results agree with our observations that the disruption of a single disulfide bond in PyEBL region 6 dramatically changes the virulence of the *P. yoelii* parasites. Only our group reported the association between parasite virulence and PyEBL localization in the 17X strain [13]. In this study, the disruption of three out of four disulfide bonds in PyEBL region 6 confirmed the association between parasite virulence and localization of PyEBL region 6.

The N-terminal Cys-rich region 2 domain mediates erythrocyte-binding of the EBL proteins, such as PvDBP, EBA-175, and PyEBL [4,17]. Therefore, a conformational change of PyEBL region 6 would not directly influence the PyEBL binding activity. Based upon structural analyses, EBA-175 region 2 was reported to form a homodimer and the dimerization is essential for the binding to the erythrocyte receptor [27]. It is possible that EBL region 6 could contribute to the dimerization of EBL proteins. Although PyEBL is essential for erythrocyte invasion and mislocalized PyEBL in the 17XL strain can be secreted onto the merozoite surface [17], it is still possible that the essential role of PyEBL is not the recognition of the erythrocyte receptor. If this is the case, mislocalized PyEBL from the micronemes might induce EBL-independent invasion pathway(s), which in turn increase parasite infectivity. The interplay between different invasion pathways has been reported in *P. falciparum*, in which disruption of the EBA-175 gene locus dramatically increased the expression of the invasion ligand reticulocyte binding-like homolog 4 (PfRh4) [28].

We found that substituting the 6th Cys (C754) to Ala or Arg was unsuccessful. A control plasmid which shared the same background sequence but does not alter the 6th Cys was successfully inserted into the target site, indicating that this failure was not due to the plasmid’s design. Substitution of the 7th Cys (C768), which is the disulfide bond partner of the 6th Cys (C754), was achieved and resulted in phenotypic changes. One possible explanation for these results is that a cysteine residue is essential at position 754 of PyEBL, even if it does not contribute to disulfide bond formation. The structural analysis of EBA-175 region 6 [19] reported that clusters of aromatic and polar residues on the protein surface might be involved in protein–protein recognition. Based on the amino acid sequence homology between EBA-175 and PyEBL region 6, C754 in PyEBL is also located in the vicinity of aromatic and polar residues, specifically, adjacent to Y753, F757, and F758. Thus, the replacement of C754 with Ala or Arg might cause a change in polarity in the recognition interface in PyEBL region 6, making the transgenic parasites non-viable. Consistent with this hypothesis is that in the *P. yoelii nigeriensis* N67 line, the 6th Cys is substituted to Tyr, an aromatic residue with a polar side chain [20]. A second possible explanation for the non-viability of a C754A line is that the free thiol side chain at the 7th Cys (C768), which is the disulfide bond partner of the 6th Cys, in isolation might cause a disruption in PyEBL function; for example, by forming a disulfide bond with another Cys and thereby impairing PyEBL region 6′s structure. Arguing against this explanation is the observation that in the *P. yoelii nigeriensis* N67 line, the 6th Cys can be substituted with Tyr [20].

Among five transgenic lines showing higher infectivity, the C717A and C768A lines are not lethal, and the C726A, C747A, and C748A lines are only partially lethal in BALB/c mice by day 14 post infection. Higher parasitemia generally kills BALB/c mice by severe anemia on days 8 or 9, and perhaps these transgenic parasites do not generate a parasite burden during the peak parasitemia as high as the lethal 17XL strain.

In summary, in PyEBL region 6 we have newly evaluated the eight Cys residues which participate in disulfide bonds which might stabilize the region 6 structure. Analysis of the successfully generated seven transgenic lines, in which one of the Cys residues in region 6 was substituted to Ala, revealed that the substitution of two Cys residues that form one disulfide bond did not change the EBL localization nor did it increase the infection rate; rather, it decreased it slightly. Substitution of the other five Cys residues that contribute to the other three disulfide bonds resulted in an alteration of the EBL localization and increased infection rate, consistent with the phenotype previously observed following the substitution of the 2nd Cys to Arg [17]. In this study we have observed the perfect association between PyEBL mislocalization and an effect on the infection course by substituting seven Cys residues in PyEBL region 6. Structural analysis of Cys-substituted EBL region 6 would shed light on a possible impact of these substitutions on the structure and possible homo- or hetero-dimer formation. Each cysteine which we substituted in this study appeared to contribute to parasite infectivity and PyEBL localization but not in a uniform manner, which may suggest complexity in the function of this domain. The intracellular trafficking of malaria invasion molecules is not well described, and so our result may help toward understanding the molecular interaction of these ligands and may possibly lead to inhibition strategies as new anti-malarial approaches. Rodent malaria parasites encode single EBL genes, and this experimental system may have potential for future studies. Similar substitutions could be explored in the human malaria parasites *P. falciparum* and *P. vivax* and their experimental systems, as well as the search for possible field isolates encoding EBL gene mutations and associations with the clinical severity of malaria. These findings may also contribute to future vaccine studies using the human malaria parasite orthologs, EBA-175 and PvDBP, by understanding the behavior of vaccine target molecules during the parasite invasion step.

## Figures and Tables

**Figure 1 biomolecules-13-00458-f001:**
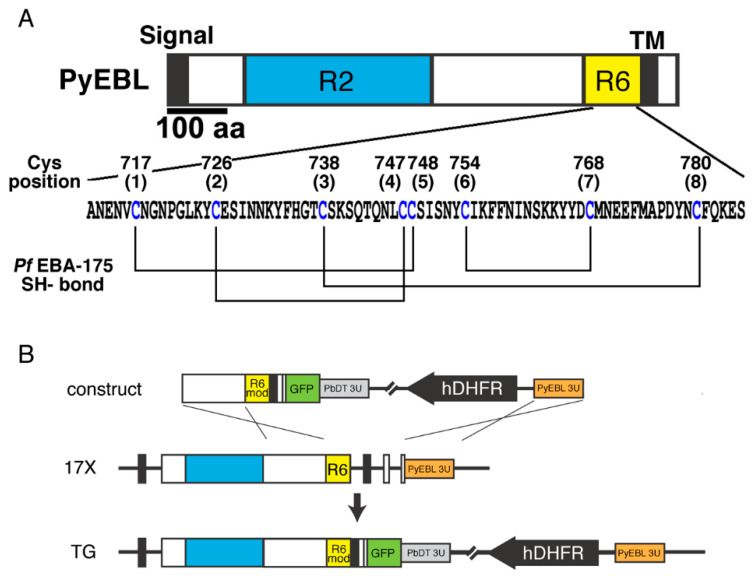
(**A**) Schematic structure of *P. yoelii* EBL (PyEBL). Signal, TM, R2, and R6 indicate the putative endoplasmic reticulum transporting signal, the transmembrane region, region 2, and region 6, respectively. The amino acid sequence of PyEBL in the 17X strain is shown below. Eight conserved Cys residues that form disulfide bonds based on the structure analysis of *P. falciparum* EBA-175 region 6 are indicated. (**B**) Schematic representation of the plasmid construct, *pyebl* gene locus in the 17X parental strain (17X), and transgenic lines (TG). The construct was inserted into the *pyebl* gene locus by double crossover recombination. In this scheme, “R6 mod” indicates the fragment of region 6 with a Cys substituted to Ala. hDHFR indicates a human dihydrofolate reductase expression cassette, which was used as a drug-resistant marker.

**Figure 2 biomolecules-13-00458-f002:**
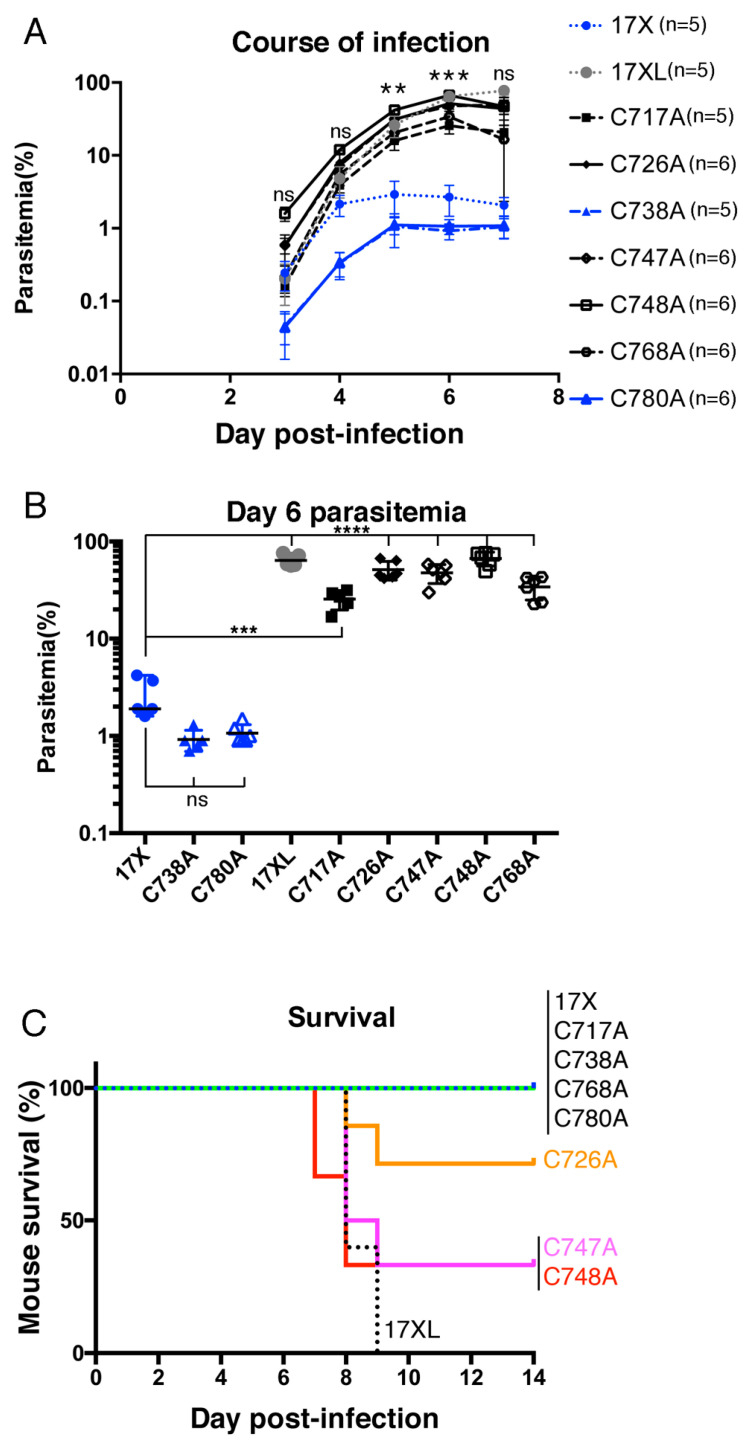
Effect of substituting Cys to Ala in PyEBL region 6 (n = 5~6 per group). Mice were intravenously inoculated with 1 × 10^4^ parasitized erythrocytes. 17X and 17XL represent the *P. yoelii* 17X and 17XL wild-type parasite strains, respectively. The names of the transgenic parasite lines are shown as an abbreviation of the position of the Cys to Ala substitution. (**A**) Infection courses in mice are presented by parasitemias from days 3 to 7. (**B**) Parasitemia on day 6. The difference was evaluated with Prism 6.0 software with a post-hoc test. (**C**) Survival curve of infected mice. All mice infected with the 17XL strain died by day 9. Some mice infected with C726A, C747A, or C748A died by day 9, but the remaining mice survived until day 14. All mice infected with other lines survived until day 14. The statistical significance of the parasitemia was tested against the 17X parental line by one-way ANOVA with Dunnett’s multiple comparison test; **, ***, and **** indicate *p* < 0.01, *p* < 0.001, and *p* < 0.0001, respectively. ns, not significant.

**Figure 3 biomolecules-13-00458-f003:**
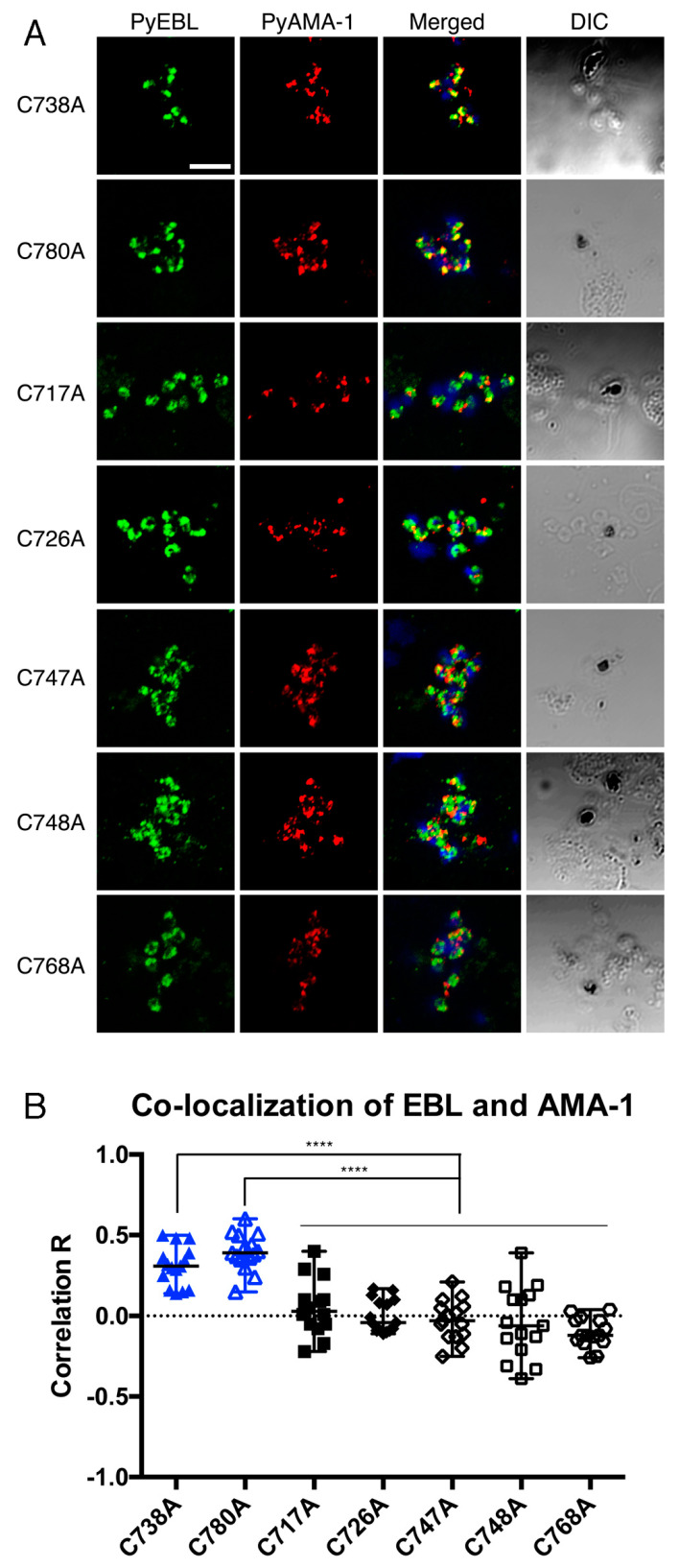
(**A**) Transgenic *P. yoelii* schizonts were labelled with mAb 1G10 (α-PyEBL) in green, rabbit anti-PyAMA-1 serum (α-PyAMA-1) in red, and Hoechst 33342 (nucleus) in blue. Differential interference contrast (DIC) images are shown on the right. Transgenic parasite lines with higher parasitemia (C717A, C726A, C747A, C748A, and C768A) displayed PyEBL signals which did not co-localize well with PyAMA-1. The other transgenic parasite lines with lower parasitemia (C738A and C780A) showed good PyEBL colocalization with PyAMA-1. The scale bar indicates 5 µm. (**B**) To quantify co-localization, PyEBL (green) and PyAMA-1 (red) signals in each image were compared by ZEN software (Zeiss). Fifteen schizont images were analyzed from each transgenic parasite line. Each dot in the scatter plot represents the correlation coefficient between PyEBL and PyAMA-1 signals in each schizont. The difference was evaluated with Prism 6.0 software with post-hoc test. **** indicates *p* < 0.0001.

**Figure 4 biomolecules-13-00458-f004:**
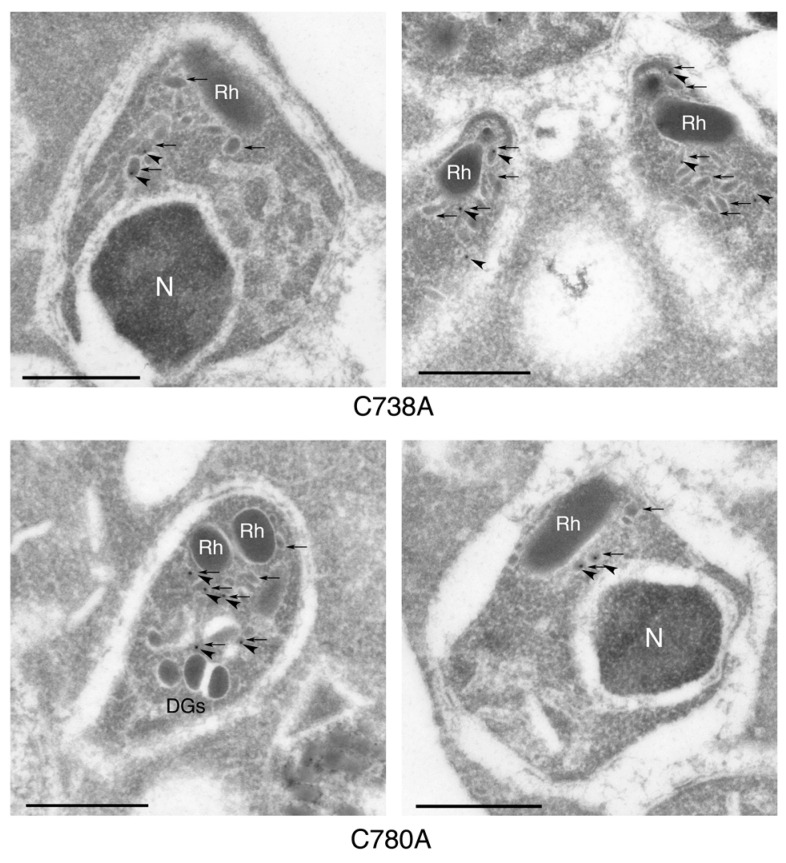
Immunoelectron microscopy of mature merozoites in schizonts obtained from *P. yoelii* transgenic parasite lines, C738A and C780A. PyEBL was probed with rat anti-PyEBL serum and a secondary antibody conjugated with gold particles. PyEBL was detected in the micronemes of each merozoite. N, DGs, and Rh indicate nucleus, dense granules, and rhoptry, respectively. Arrowheads and arrows indicate gold particles and micronemes, respectively. The scale bars indicate 500 nm.

**Table 1 biomolecules-13-00458-t001:** Selectivity index of wild-type and transgenic parasite lines.

Parasite	Substituted Cys	Selectivity Index (Range)
C717A	Cys^1st^ to Ala	7.3 (4.8–9.4)
C726A	Cys^2nd^ to Ala	1.1 (0.8–1.6)
C738A	Cys^3rd^ to Ala	46.8 (41.2–51.5)
C747A	Cys^4th^ to Ala	2.1 (1.6–2.8)
C748A	Cys^5th^ to Ala	0.5 (0.5–0.6)
C768A	Cys^7th^ to Ala	7.3 (6.4–8.2)
C780A	Cys^8th^ to Ala	37.3 (16.4–61.3)
17X (parental non-lethal strain)		14.6 (11.3–17.0)
17XL (lethal strain)	Cys^2nd^ to Arg	6.8 (3.4–10.4)

## Data Availability

All data are contained within the article.

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
