# Peer review of "Cysteine Residues in Region 6 of the Plasmodium yoelii Erythrocyte-Binding-like Ligand That Are Related to Its Localization and the Course of Infection"

_biomolecules, 2023, doi:10.3390/biom13030458_

Round 1

Reviewer 2 Report

In this manuscript (Biomolecules-2204456), authors have established 8 transgenic P. yoelii parasite lines, specifically, C738A and C780A, do not help in localization of protein and decreased their infectivity as well; rather, they grew slower than the parental 17X strain. Conclusively, results suggest that the disulfide bond between Cys residues at amino acid positions 738 (3rd Cys) and 780 (8th Cys) is not essential for the trafficking of PyEBL to the micronemes.

Authors has done nice work as they have done in the past similar kind work on set pattern. Submitted manuscript is also well written though, there are some minor suggestions.

It would be more convincing, if genomic architecture of the region with the primer set can be incorporated in the figure 1. You can find this kind of pictorial presentation in this paper figure 1. https://doi.org/10.1016/j.gene.2016.09.022

Figure 2A x-axis should be started from “0” which is considered a day of injection and resultantly initial time-point data would be more visible as shown in reference 17.

If figure 2A and 2B are sharing the same symbol for each transgenic line, then 17XL is not showing the same representation.

It would be pleasant to readers, if you keep the same symbol in graphs for each transgenic lines between the figures too.

Overall, paper has well-written but recheck the language and other minor mistake at your end.

All the best.

Author Response

COMMENT: 1) It would be more convincing, if genomic architecture of the region with the primer set can be incorporated in the figure 1. You can find this kind of pictorial presentation in this paper figure 1. https://doi.org/10.1016/j.gene.2016.09.022

RESPONSE: We tried to integrate primer information into Figure 1, but we amplified gene fragments from both genomic DNA and cDNA and we found that it would be difficult to show in one figure.

COMMENT: 2) Figure 2A x-axis should be started from “0” which is considered a day of injection and resultantly initial time- point data would be more visible as shown in reference 17.

RESPONSE: We accordingly modified x-axis of Figure 2A.

COMMENT: 3) If figure 2A and 2B are sharing the same symbol for each transgenic line, then 17XL is not showing the same representation.

RESPONSE: The symbols were accordingly unified to avoid misunderstandings.

COMMENT: 4) It would be pleasant to readers, if you keep the same symbol in graphs for each transgenic lines between the figures too.

RESPONSE: We unified all symbols to be correspondent to each transgenic parasite line.

COMMENT: 5) Overall, paper has well-written but recheck the language and other minor mistake at your end.

RESPONSE: Thank you for your comment. We carefully rechecked the language of our manuscript with a malaria expert English editor, Dr. Thomas J. Templeton. 

Reviewer 3 Report

Excellent manuscript, thank you for this contribution to the literature.

Author Response

COMMENT: Excellent manuscript, thank you for this contribution to the literature.

RESPONSE: Thank you very much for your encouragement.

Reviewer 4 Report

After reading the manuscript entitled "Cysteine residues in the region 6 of the Plasmodium yoelii erythrocyte-binding-like ligand that are related to its localization and the course of infection", I would like to make a few comments.

The main remark concerns the "Introduction" section. In my opinion, this section could be slightly rewritten and supplemented, because:

 1)  the authors mention several species of Plasmodium without giving a preliminary explanation of their epidemiological significance;

2  the authors mention Duffy antigen without first explaining what it is

Less important notes:

1) Figure 2A Strain designations are merging, which is not very convenient for the perception of the figure;

2) I would like to ask - Is it possible to extrapolate the results obtained by the authors to Plasmodium pathogenic for humans? Maybe it's worth adding such reasoning to the "discussion" section?

Author Response

COMMENT: 1) the authors mention several species of Plasmodium without giving a preliminary explanation of their epidemiological significance;

RESPONSE: In accordance with your suggestion, we added explanation sentence in introduction section (lines 41-44).

COMMENT: 2) the authors mention Duffy antigen without first explaining what it is.

RESPONSE: We accordingly added a sentence in the introduction section (lines 80-81).

Less important notes:

COMMENT: 1) Figure 2A Strain designations are merging, which is not very convenient for the perception of the figure;

RESPONSE: We apologize that the parasitemias of each line were so close. To improve this, we separately showed Day 6 parasitemia in Fig. 2B, for a more spaced comparison of each line of parasites.

COMMENT: 2) I would like to ask - Is it possible to extrapolate the results obtained by the authors to Plasmodium pathogenic for humans? Maybe it's worth adding such reasoning to the "discussion" section?

RESPONSE: This is an interesting point; but, to date we could not find a similar substitution in a human malaria parasite. We hope to see what will happen if this kind of change occur in human case. We have added one last sentence in the Discussion section (lines 680-682).

Round 2

Reviewer 1 Report

While we truly appreciate many of the changes made by the authors, we feel that our major concerns were insufficiently or not addressed. MDPI Biomolecules “encourages scientists to publish their experimental results in as much detail as possible”. We felt frustrated seeing the authors have not revised the manuscript in such a way that others now are able to repeat the experiments; information is still lacking, unclear or readers are expected to look for details in other (non-open access) articles. Furthermore, conditions of the animals used in the study were still described poorly, and requested information about the ethical guidelines was not provided. Most importantly, the authors did not address our concern that this manuscript is extremely similar to the one published in PNAS in 2009. The authors failed to show us what new insight this work brings to the scientific community. We felt studying and describing if any of the disulfide-disrupting cysteine substitutions in EBL proteins could be found in circulating P. falciparum or P. vivax strains would make this manuscript more interesting, with a novel finding, but the authors decided to not do this. Based on all concerns described above, we unfortunately cannot support the publication of this work in Biomolecules.
